# Bioactive Flavonoids Icaritin and Icariin Protect against Cerebral Ischemia–Reperfusion-Associated Apoptosis and Extracellular Matrix Accumulation in an Ischemic Stroke Mouse Model

**DOI:** 10.3390/biomedicines9111719

**Published:** 2021-11-19

**Authors:** Cheng-Tien Wu, Man-Chih Chen, Shing-Hwa Liu, Ting-Hua Yang, Lin-Hwa Long, Siao-Syun Guan, Chang-Mu Chen

**Affiliations:** 1Department of Nutrition, China Medical University, Taichung 406040, Taiwan; ct-wu@mail.cmu.edu.tw; 2Master Program for Food and Drug Safety, China Medical University, Taichung 406040, Taiwan; 3Institute of Toxicology, College of Medicine, National Taiwan University, Taipei 10051, Taiwan; r06447007@ntu.edu.tw (M.-C.C.); shinghwaliu@ntu.edu.tw (S.-H.L.); 4Department of Medical Research, China Medical University Hospital, China Medical University, Taichung 406040, Taiwan; 5Department of Pediatrics, College of Medicine and Hospital, National Taiwan University, Taipei 10051, Taiwan; 6Department of Otolaryngology, National Taiwan University Hospital, Taipei 10051, Taiwan; thyang37@ntu.edu.tw; 7Division of Neurosurgery, Department of Surgery, College of Medicine and Hospital, National Taiwan University, Taipei 10051, Taiwan; linhua1976@yahoo.com.tw; 8Institute of Nuclear Energy Research, Atomic Energy Council, Taoyuan 32546, Taiwan; ssguan@iner.gov.tw

**Keywords:** extracellular matrix, icariin, icaritin, ischemic stroke, middle cerebral artery occlusion

## Abstract

Stroke, which is the second leading cause of mortality in the world, is urgently needed to explore the medical strategies for ischemic stroke treatment. Both icariin (ICA) and icaritin (ICT) are the major active flavonoids extracted from *Herba epimedii* that have been regarded as the neuroprotective agents in disease models. In this study, we aimed to investigate and compare the neuroprotective effects of ICA and ICT in a middle cerebral artery occlusion (MCAO) mouse model. Male ICR mice were pretreated with both ICA and ICT, which ameliorated body weight loss, neurological injury, infarct volume, and pathological change in acute ischemic stroke mice. Furthermore, administration of both ICA and ICT could also protect against neuronal cell apoptotic death, oxidative and nitrosative stress, lipid peroxidation, and extracellular matrix (ECM) accumulation in the brains. The neuroprotective effects of ICT are slightly better than that of ICA in acute cerebral ischemic stroke mice. These results suggest that pretreatment with both ICA and ICT improves the neuronal cell apoptosis and responses of oxidative/nitrosative stress and counteracts the ECM accumulation in the brains of acute cerebral ischemic stroke mice. Both ICA and ICT treatment may serve as a useful therapeutic strategy for acute ischemic stroke.

## 1. Introduction

According to the statistic report of the World Health Organization (WHO), stroke is the second leading cause of mortality in the world [1]. The health care disbursements spent on stroke have been estimated to be approximately 3% to 4% of total expenditures in many countries [2]. Although the death rates and prevalence of stroke have reduced over time, the medical burden or substantially rehabilitative economic costs for post-stroke care remain likely to become a heavy load of many countries [3]. The latest published Global Burden of Disease Study revealed an estimated 5.5 million people died due to stroke, while 116 million people suffered stroke-associated problems, including aphasia, spasticity, and memory problems worldwide [4]. On the basis of the difference of the etiological process of stroke, approximately 80% of all cases are ischemic stroke, the rest being hemorrhagic stroke [5]. Ischemic stroke occurs when clogged blood vessels occur due to thrombus or atherosclerosis, leading to the interruption of the blood supply to the brain [6]. During ischemic stroke, low levels of oxygen are delivered into the brain to first evoke the hypoxia response signals, and then the reperfusion injury-associated networks, such as glutamate excitation with calcium overload, oxido-nitrosative stress, danger-associated molecular patterns (DAMPS) [7], matricellular protein accumulation [8,9], and proinflammatory induction [10,11], which are aroused after removing or dissolving the clots. The increased number of peri-infarct depolarization during ischemic stroke triggers a larger infarct region [12]. The apoptotic neuronal cell death [13] and accumulated fibrotic proteins [14] are consequently developed to lead the permanent brain damage. The therapeutic effects of traditional thrombolytic agents, including plasminogen activators, for ischemic stroke are limited. Exploring the medical strategies for ischemic stroke prevention or therapy is urgently needed. 

Both icariin (ICA) and icaritin (ICT) are the major bioactive flavonoids isolated from the Chinese medicine horny goat weed (also known as Ying Yang Huo and *Herba epimedii*) [15]. ICT, a metabolite of ICA, has been suggested to process several biological activities, including neuroprotection against β-amyloid-induced neurotoxicity [16], immunomodulation [17], and anticancer effects [18]. ICA and ICT have been used to improve memory and learning abilities in experimental Alzheimer’s disease models [19,20]. Furthermore, ICT has also been indicated to reveal an anti-inflammatory property and adjust a chloride influx in mouse brain cortical cells [21] or act as an antioxidant agent to prevent neuronal cells against oxidative stress [22]. Zhu et al. [23] and Xiong et al. [24] have shown that ICA treatment can protect the brain from ischemia–reperfusion injury in mice and rats, respectively. Recently, the neuroprotective effects of ICT on focal cerebral ischemic–reperfusion injury in mice have been reported [17]. However, the detailed effects and mechanisms of both ICA and ICT on acute cerebral ischemic stroke still remain to be clarified.

In this study, we aimed to investigate and compare the effects and mechanisms of ICA and ICT on acute ischemic stroke using a middle cerebral artery occlusion (MCAO) mouse model. The neuronal functions, infarct volume, and pathology changes were assessed. The examinations for apoptosis, oxidative/nitrosative stress, and matricellular proteins in the cortex and hippocampus were also performed.

## 2. Materials and Methods

### 2.1. Cerebral Ischemia–Reperfusion (I/R) Injury Model

Male ICR mice aged 4 to 5 weeks old were purchased from the Laboratory Animal Center of the College of Medicine, National Taiwan University (Taipei, Taiwan). Mice were housed under controlled temperature as well as photoperiod conditions (12 h light/dark) with food and water freely available. All animal surgical procedures were followed to the approved animal protocol (no. 20190111) and guidelines of the Animal Research Committee of the College of Medicine in National Taiwan University. Briefly, mice (*n* = 30) were randomly divided into five groups, namely, sham, I/R, I/R + edaravone (3 mg/kg; Selleck Chemicals, Houston, TX, USA), I/R + ICA (60 mg/kg), and I/R + ICT (60 mg/kg). Drugs were dissolved in dimethyl sulfoxide (DMSO; Sigma-Aldrich, St. Louis, MO, USA). The dosage of 60 mg/kg for both ICA and ICT was selected according to the previous studies [25,26] and our preliminary test. Edaravone treatment was as a positive control [27]. ICA and ICT were purchased from Sigma-Aldrich (cat. no.: 56601, Purity > 95%; St. Louis, MO, USA) and Cayman Chemical (cat. no.: Cay20236-500, Purity > 98%; Ann Arbor, MI, USA), respectively. Drugs edaravone, ICA, and ICT were given by intraperitoneal injection (i.p.) before the focal cerebral ischemia for 1 h, and then the procedure of MCAO surgery was carried out. Mice were anesthetized by inhalation of isoflurane (Tokyo Chemical Industry Co., Tokyo, Japan), which was mixed with 3% oxygen. The middle incision was operated on the neck, and then the left common carotid artery was separated. A 6-0 nylon thread was inserted from the incision of external carotid artery to the middle cerebral artery for 50 min occlusion. During this process, a Laser Doppler (PeriFlux 4001, Perimed, Stockholm, Sweden) was applied to monitor the cerebral blood flow of MCAO mice as previously described [28,29]. The same procedure was also operated on the sham group without the nylon thread insertion. After 24 h of reperfusion, all mice were euthanized, and necropsy was performed to observe the ischemic stroke signal networks. The rectal temperature of mice was maintained at 37.0 °C with a temperature-control heating pad during and after the MCAO surgery. The dose selection for ICT, ICA, and edaravone was decided according to the previous studies and a preliminary test.

### 2.2. Neurological Score Assessment

To evaluate the neurological injury, we assessed the modified neurological severity score (mNSS). It was scored on a scale of 0 to 14 (normal score, 0; maximal deficit score, 14), which was detected for the behavior of motor, sensory, reflex, and balance. The evaluated neurological functions for mice have been indicated and modified by Li et al. [30]. The scales were followed to the levels: 10–14, severe injury; 6–10, moderate injury; 1–5, mild injury.

### 2.3. Determination of Infarct Volume and Histopathological Detection

After assessing the behavior and neuronal functions, we euthanized the mice, and the brains were isolated and sliced into 2 mm thick coronal sections. The tissue sections were stained with 2% 2,3,5-triphenyl tetrazolium chloride (TTC; Sigma-Aldrich, St. Louis, MO, USA) at 37 °C for 20 min. In viable brain tissue, TTC was converted by mitochondria to appear red in color, while the colorless area was considered as an infarct. TTC-stained slices were photographed, and infarct volumes were analyzed by ImageJ software [31]. The infarct areas were summed and divided through the total volume of the slices, which were shown as the percentage of the volume of the contralateral hemisphere. 

The brain tissue preparation and histopathological detection were determined as previously described [29]. The paraffin-embedded 4 µm sections were stained with hematoxylin and eosin (H&E; Sigma-Aldrich, St. Louis, MO, USA) for histological examination. 

### 2.4. Terminal Deoxynucleotidyl Transferase (TdT) dUTP Nick end Labeling (TUNEL) Assay

The TUNEL assay was determined by a DeadEnd™ Fluorometric TUNEL System (Promega, Madison, WI, USA) as previously described [30]. The procedure was followed to the manufacturer’s instruction to detect the DNA fragments of late apoptotic cells. Briefly, brain slides were deparaffinized at 60 °C for 30 min and then transferred to xylene buffer for washing and then rehydrated through the decreased strength of ethanol/saline buffer with 0.85% NaCl. The sections were fixed with paraformaldehyde for 15 min. The slides were washed with a saline buffer and incubated in a dark humidity chamber at 37 °C in 100 μL of TdT incubation buffer for 1 h. The sections were counterstained with 4′,6-diamidino-2-phenylindole (DAPI; Sigma-Aldrich, St. Louis, MO, USA), slides were mounted, and we detected the fluorescein-12-dUTP-label DNA by using a fluorescence microscope.

### 2.5. Western Blotting Analysis

The brain tissues of the cortex and hippocampus were collected. The samples were lysed by radioimmunoprecipitation (RIPA; Sigma-Aldrich, St. Louis, MO, USA) buffer and centrifuged at 13,000 rpm for 30 min. After the quantification of protein with a bicinchoninic acid (BCA) protein assay kit (Thermo Fisher Scientific, Waltham, MA, USA), the equal concentrations (10–20 μg) of supernatants with sodium dodecyl sulfate (SDS; Millipore, Burlington, MA, USA) buffer were heated at 95 °C for 10 min. Next, the protein samples were separated by 10–15% SDS-polyacrylamide gel electrophoresis (SDS-PAGE) and blotted onto polyvinylidene difluoride (PVDF) membrane (Millipore, Burlington, MA, USA). The membranes were blocked with 5% skimmed milk, which was dissolved in 0.1% TBST (50 mM Tris-HCl (pH 7.5), 150 mM NaCl, 0.1% Tween 20) buffer for 1 h. Samples were incubated with primary antibodies for CD31 (#3528, 1:1000 dilution; Cell Signaling, Danvers, MA, USA), Bax (#14796, 1:1000 dilution; Cell Signaling, Danvers, MA, USA), Bcl-2 (#3498, 1:1000 dilution; Cell Signaling, USA), caveolin-1 (#3267, 1:1000 dilution; Cell Signaling, Danvers, MA, USA), cleaved caspase 3 (#9664, 1:1000 dilution, Cell Signaling, Danvers, MA, USA), cleaved PARP (#9548, 1:1000 dilution, Cell Signaling, Danvers, MA, USA), vimentin (#5741, 1:1000 dilution, Cell Signaling, Danvers, MA, USA), eNOS (#32027, 1:1000 dilution, Cell Signaling, USA), fibronectin (#26836, 1:1000 dilution, Cell Signaling, Danvers, MA, USA), iNOS (610431, 1:1000 dilution, BD Biosciences, San Jose, CA, USA), and β-actin (sc-47778, Santa-Cruz, Dallas, TX, USA) overnight at 4 °C. Samples were washed by 1% TBST for 10 min three times and then incubated with HRP-conjugated secondary antibodies for 1 h. Finally, the levels of protein expression were densitometric quantification by ImageJ analysis software [32] and normalized by β-actin.

### 2.6. Measurement of Lipid Peroxide (Thiobarbituric Acid Reactive Substances, TBARS) Levels

The levels of malondialdehyde (MDA; a product of lipid peroxidation) were measured as previously described [33]. A TBARS Assay Kit (Cayman Chemical, Ann Arbor, MI, USA) for colorimetric measurement of MDA was used. The brain tissues of the cortex and hippocampus were collected. The reaction between thiobarbituric acid (TBA) and MDA in the tissue homogenates was performed. The absorbance at 540 nm was detected by a spectrophotometer.

### 2.7. Statistical Analysis

The results are presented as the mean ± standard error of the mean of at least three independent experiments. The statistical significance of differences between groups was analyzed by one-way analysis of variance (ANOVA) and followed by Dunnett’s post hoc test. When the *p*-value was less than 0.05, it was considered as a significant difference. Data analysis was performed by the GraphPad Prism software (San Diego, CA, USA).

## 3. Results

### 3.1. Both ICT and ICA Ameliorated the Neurological Functions and Brain Pathological Changes in Acute Ischemic Stroke Mice

To evaluate the neuroprotective potency, we pretreated ICA, ICT, and edaravone (a positive control) before MCAO surgery in mice. The chemical structures of ICA and ICT are shown in Figure 1A. We first observed the effects of these drugs on the brain pathological changes and neuronal function loss in MCAO mice. As shown in Figure 1B–E, the bodyweight loss, infarct volume, and the mNSS assessment were significantly increased in MCAO group. After both ICA and ICT treatment, these pathological changes and neurological dysfunctions were significantly reversed. Edaravone treatment significantly reversed the pathological changes and neurological dysfunctions, but not bodyweight loss, in MCAO mice (Figure 1). The histopathological changes of the cerebral cortex and hippocampus in MCAO mice showed a locally extensive neuronal necrosis, neuronal cell loss with cytoplasmic vacuolation of neuropils, and the increased amount of irregularly atrophic neuronal cells as well as the shrunken nucleus (Figure 2). The hemorrhagic dots were also presented in the brain cortex after MCAO surgery (Figure 2). These histopathological changes in the cerebral cortex and hippocampus of MCAO mice could be effectively reversed by both ICA and ICT treatment (Figure 2).

### 3.2. Both ICA and ICT Protected against Neuronal Cell Apoptosis in the Brains of Acute Ischemic Stroke Mice

To evaluate the numbers of apoptotic cells in the ischemic brain, we performed TUNEL staining to detect the fragmented DNA of apoptotic cells. As shown in Figure 3, the TUNEL-positive cells were revealed as a green color while the cell nucleus was stained by DAPI to a blue color. TUNEL-positive cells were clearly observed in the cerebral hippocampus and cortex of MCAO mice, which could be effectively reversed by both ICA and ICT treatment. 

We next examined the protein expression for apoptosis-related signaling molecules in the cerebral hippocampus and cortex as determined by Western blot. As shown in Figure 4, the levels of protein expression for cleaved caspase-3, cleaved PARP, and Bax were markedly increased, while the protein expression of Bcl-2 was dramatically decreased in the MCAO group. Nonetheless, pretreatment with both ICA and ICT could effectively reverse the changes of protein expression for these apoptosis-related proteins in MCAO mice (Figure 4).

### 3.3. Both ICT and ICA Counteracted Oxidative Stress and Nitrosative Stress in the Brains of Acute Ischemic Stroke Mice 

We next observed the changes in the levels of protein expression for antioxidant enzymes and nitric oxide synthases and the lipid peroxide generation in acute ischemic stroke mice. As shown in Figure 5A, the levels of protein expression of catalase and SOD-1 were significantly decreased in the cortex and hippocampus of MCAO mice. Both ICT and ICA treatment significantly and conspicuously reversed the decreased protein expression of SOD-1, while partially reversing the decreased catalase protein expression (Figure 5A). 

The activated nitrosative stress is one of the important initial signals to arouse the stroke-induced neuroinflammation and the followed extracellular matrix deposition as well as fibrosis [8]. We next investigated whether both ICA and ICT possess preventive effects on the reduction of nitrosative stress. Caveolin-1 can react with NOSs and inhibit NO synthesis [34]. Caveolin-1 has been found to be reduced after cerebral ischemia–reperfusion injury [35]. As shown in Figure 5A, the protein expression of caveolin-1 was diminished after cerebral ischemia–reperfusion injury, while both eNOS and iNOS protein expression was elevated. Pretreatment with ICT effectively reversed the changes in these protein expression levels in the cortex and hippocampus of MCAO mice (Figure 5A). Moreover, the levels of lipid peroxidation product MDA in the brains of MCAO mice were markedly and significantly increased in MCAO mice, which could be reversed by both ICA and ICT administration (Figure 5B). 

### 3.4. Both ICT and ICA Alleviated the Endothelial–Mesenchymal Transition in the Ischemic Stroke Mice

It has been reported that ischemia–reperfusion may lead to endothelial cell damage and trigger endothelial-to-mesenchymal transition (EndMT) during ischemic acute kidney injury [36]. We next examined the effects of ICA and ICT on endothelial–mesenchymal transition in the brains of MCAO mice. As shown in Figure 6, the levels of protein expression for CD31 (an endothelial cell marker) as well as fibronectin and vimentin (the fibroblast/mesenchymal markers) were significantly decreased and increased, respectively, in the brains of MCAO mice. Both ICA and ICT treatment significantly reversed the increased fibronectin and vimentin protein expression in both cortex and hippocampus tissues of MCAO mice (Figure 6). Both ICA and ICT treatment could not significantly reverse the decreased CD31 protein expression in the cortex tissues of MCAO mice, but ICT treatment significantly reversed the decreased CD31 protein expression in the hippocampus tissues of MCAO mice (Figure 6). These results suggest that both ICA and ICT can inhibit endothelial–mesenchymal transition and extracellular matrix accumulation in the brains of acute ischemic stroke mice. 

## 4. Discussion

Stroke is a life-threatening morbidity condition that causes long-term disability. Although the first FDA-approved thrombolytic agent recombinant tissue plasminogen activator for stroke has been developed for a decade, the current therapeutic drugs remain problematic and controversial. Researchers have developed the therapeutic strategies for stroke including searching the neuroprotective agents, such as antioxidants, anti-inflammatory agents, and anti-atherosclerosis drugs [37,38,39]. Both ICA and ICT, the bioactive compounds from *Herba epimedii*, have been revealed to possess the biological properties of antioxidant and anti-atherosclerosis [40,41]. ICA administered by gavage at doses of 50–200 mg/kg/day after reperfusion has been shown to alleviate ischemia reperfusion-induced brain injury in MCAO mice [23]. Xiong et al. [24] have also found that ICA administered by gavage at doses of 10 and 30 mg/kg twice per day for three consecutive days before reperfusion attenuates cerebral ischemia–reperfusion injury. Recently, ICT administered by intraperitoneal injection at a dose of 3 mg/kg/day at before reperfusion has been shown to possess the neuroprotective effects in cerebral I/R mice [17]. In this study, our results revealed that both ICA and ICT administered by intraperitoneal injection at a dose of 60 mg/kg effectively improved brain injury in acute cerebral ischemic stroke mice. The efficacy of ICT seemed to be slightly better than that ICA treatment. In a cell model of Alzheimer’s disease, the effects of ICT on decreasing the levels of GSK-3β and phosphorylated Tau have been found to be slightly better than that of ICA [19]. Taken together, these results suggest that both ICA and ICT treatment may potentially possess the advantaged neuroprotective function in stroke. 

The progression of ischemic stroke can induce the activation of stress signals, hypoxia, oxygen–glucose deprivation, and oxidative/nitrosative stress, leading to further neuro-inflammation and fibrosis in the brains [8]. As the stroke clot is removed, the reperfusion injury may refer to the reactive oxygen species (ROS)/reactive nitrogen species (RNS) response; protein expression of EMC; and its associated proteins such as matrix metalloproteinases [42,43], basement membrane changes [44], and inflammation induction [45]. Sun et al. [17] have indicated that ICT pretreatment effectively prevents the neuroinflammatory response and oxidative damage in the brains of cerebral ischemia–reperfusion mice. On the other hand, cerebral ischemic stroke-induced apoptosis is known to contribute to a significant proportion of neuronal death [13,46]. The overproduction of free radicals, Ca^2+^ overload, and excitotoxicity may be the key molecular events to initiate apoptosis during acute brain ischemia [46]. Both apoptotic and anti-apoptotic proteins have been suggested to be simultaneously over-expressed in the penumbra after ischemic stroke [47]. In the present study, we observed that treatment with both ICA and ICT could also mitigate the pathophysiological changes, TUNEL-positive neuronal cells, imbalance of pro-apoptotic and anti-apoptotic proteins, apoptosis-related signaling molecules, and ROS/RNS-related signaling molecules in the hippocampus and cortex of the MCAO mice. Taken together, these results suggest that both ICA and ICT treatment protects against ischemic stroke injury-associated oxidative/nitrosative stress and apoptosis in the brain. 

Fibrosis refers to excess accumulation of fibrous connective tissue during the repair process reacted to specific damage in tissues. EndMT, which endothelial cells lose in their specific phenotype and de-differentiate into cells with mesenchymal phenotype, is a complex process involved in physiologically embryonic development and pathogenesis of human diseases, such as vascular, inflammatory, and fibrotic disorders and cancer [48,49]. EndMT-derived fibroblasts may contribute to the formation of atherosclerotic plaques, leading to the development of cardiac or vascular disorders including stroke [49]. EndMT has been shown to be induced in an acute renal ischemia–reperfusion pig model that the expression of endothelial marker CD31 was significantly decreased, while the expression of fibrotic marker α-SMA was obviously elevated after 24 h reperfusion [50]. Jiang et al. [51] have revealed that vimentin participates in neurotoxicity and microglia activation in cerebral ischemia mice. Fasipe et al. [52] have indicated that the pharmacological targets to the vimentin/VWF (von Willebrand Factor) interaction complex can effectively improve brain injury after ischemic stroke. Furthermore, it has been indicated that the expression of fibronectin is associated with brain edema, hemorrhagic transformation, and poor functional outcome after stroke [53]. Khan et al. [54] have also found that the expression of fibronectin promotes inflammatory injury after ischemic stroke, which prolongs chronic inflammatory conditions. In the present study, we also found that endothelial marker CD31 was decreased, and fibroblastic/mesenchymal markers fibronectin and vimentin were increased in the brains of acute cerebral ischemia–reperfusion mice, which could be effectively reversed by administration of both ICA and ICT. These results suggest that both ICA and ICT administration counteract the EndMT induction and improves ischemic stroke injury in mice.

## 5. Conclusions

In conclusion, these results demonstrate for the first time that both ICA and ICT pretreatment ameliorate the acute cerebral ischemia–reperfusion injury through the improvement in apoptotic neuronal cell death, ROS/RNS-induced lipid peroxidation, ECM accumulation, and EndMT-related fibrosis in the mouse brains (Figure 6B). Both icariin and icaritin treatment may serve as a useful therapeutic strategy for acute ischemic stroke.

## Figures and Tables

**Figure 1 biomedicines-09-01719-f001:**
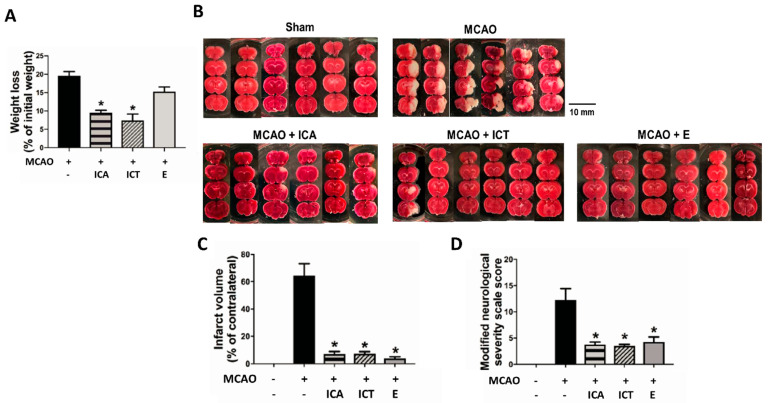
The preventive effects of both icariin (ICA) and icaritin (ICT) on body weight loss, neurological severity, and infarct volume in mice with acute cerebral ischemia–reperfusion. Mice were pretreated with ICA (60 mg/kg), ICT (60 mg/kg), and edaravone (E; 3 mg/kg, as a positive control) before MCAO followed by reperfusion. (**A**) The average body weight loss percentage of mice in each group was calculated. (**B**) Photographs of the mice cerebral infarct areas in each group were shown. (**C**) The quantification of infarct volume is shown. (**D**) The modified neurological severity score (mNSS) in each group was evaluated. Data are presented as mean ± SD (*n* = 6). * *p* < 0.05 compared to the MCAO group.

**Figure 2 biomedicines-09-01719-f002:**
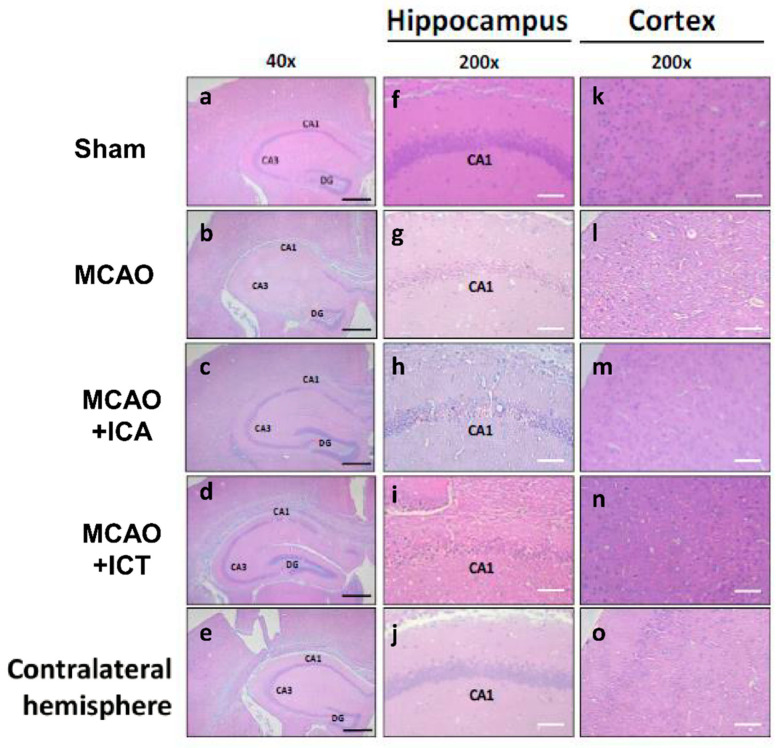
Both ICA and ICT treatment alleviated the histological changes in the left cerebral hemisphere of mice with acute cerebral ischemia–reperfusion. H&E staining was used to detect the histological changes of both cortex and hippocampus under 40× and 200× magnification. The typical ischemia–reperfusion-injured neuronal cells are demonstrated to be irregularly atrophic and eosinophilic cytoplasm as well as a shrunken nucleus with darkly stained pyknotic nuclei (irreversible condensation of chromatin). Black scale bar = 100 μm; white scale bar = 500 μm. Sham (**a**,**f**,**k**), MCAO (**b**,**g**,**l**), MCAO + ICA (**c**,**h**,**m**), MCAO + ICT (**d**,**i**,**n**), and contralateral hemisphere (**e**,**j**,**o**) are shown. The results are representative from at least four independent experiments.

**Figure 3 biomedicines-09-01719-f003:**
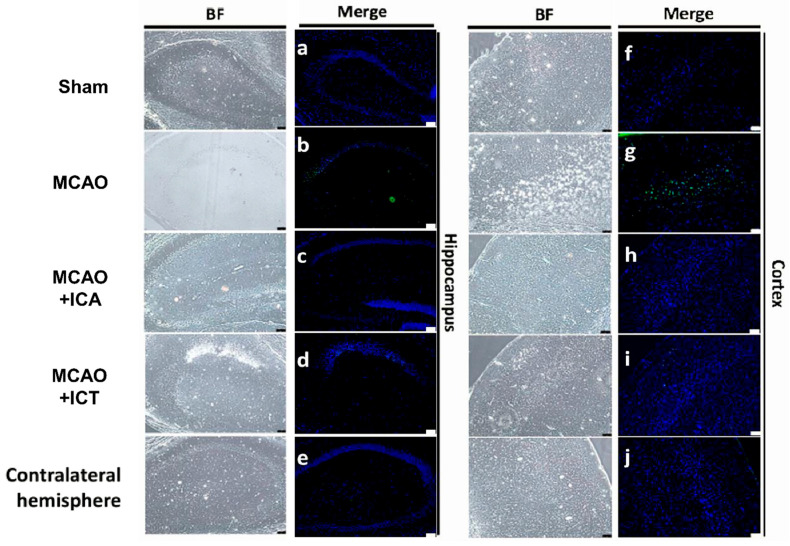
Both ICA and ICT treatment inhibited the neuronal cell apoptosis in the left cerebral hemisphere of mice with acute cerebral ischemia–reperfusion. TUNEL staining was used to detect neuronal apoptotic cells in the hippocampus ((**a**), Sham; (**b**), MCAO; (**c**), MCAO + ICA; (**d**), MCAO + ICT; (**e**), Contralateral hemisphere) and cortex ((**f**), Sham; (**g**), MCAO; (**h**), MCAO + ICA; (**i**), MCAO + ICT; (**j**), Contralateral hemisphere). TUNEL-positive cells were presented as a fluorescent green color, while cell nuclei were displayed as a fluorescent blue color. Scale bar = 75 μm. BF: bright field. The results are representative of at least four independent experiments.

**Figure 4 biomedicines-09-01719-f004:**
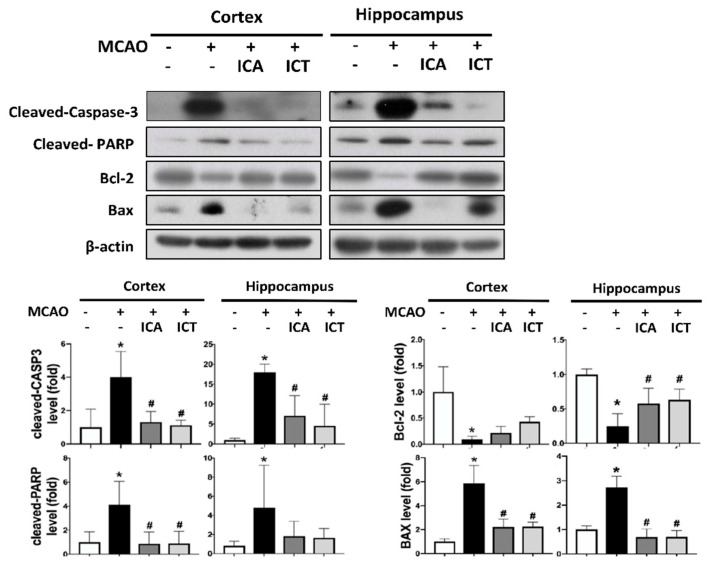
Both ICA and ICT treatment reduced the levels of protein expression of apoptotic markers in the left cerebral hemisphere of mice with acute cerebral ischemia–reperfusion. The levels of protein expression of apoptotic markers (cleaved caspase-3, cleaved PARP, Bcl-2, and Bax) in the hippocampus and cortex areas were determined by Western blot. The quantification of protein expression was determined by densitometry. Data are presented as mean ± SD (*n* ≥ 4). * *p* < 0.05 compared to sham group; # *p* < 0.05 compared to MCAO group.

**Figure 5 biomedicines-09-01719-f005:**
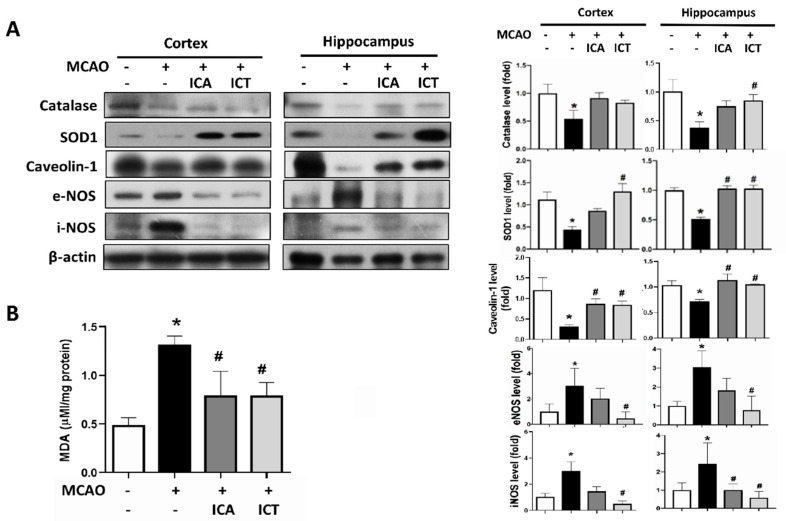
Both ICA and ICT treatment decreased oxidative/nitrosative stress and lipid peroxidation in the left cerebral hemisphere tissues of mice with acute cerebral ischemia–reperfusion. (**A**) The levels of protein expression for antioxidant enzymes (catalase and SOD1) and nitrosative stress-related proteins (caveolin-1, eNOS, and iNOS) in the hippocampus and cortex tissues were determined by Western blot. The quantification of protein expression was determined by densitometry. (**B**) The measurement of brain lipid peroxidation product MDA is shown. Data are presented as mean ± SD (*n* ≥ 4). * *p* < 0.05 compared to sham group; # *p* < 0.05 compared to MCAO group.

**Figure 6 biomedicines-09-01719-f006:**
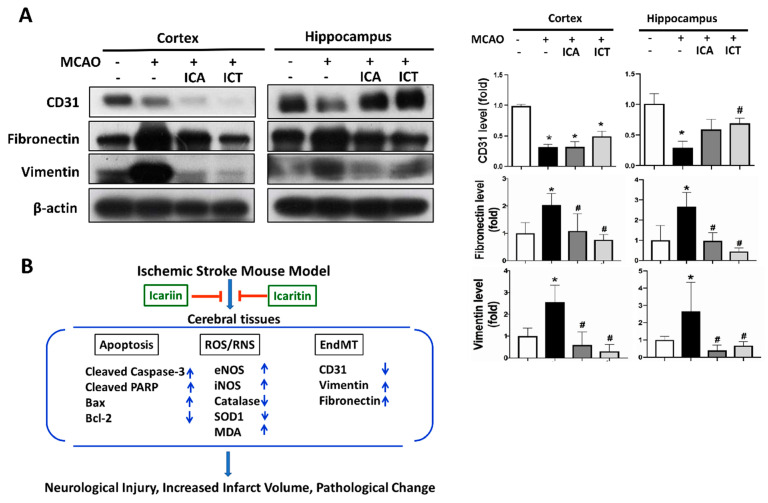
Both ICA and ICT treatment reduced endothelial-to-mesenchymal transition and extracellular matrix accumulation in the left cerebral hemisphere tissues of mice with acute cerebral ischemia–reperfusion. (**A**) The levels of protein expression for CD31 (endothelial marker) and fibronectin and vimentin (fibroblast markers) in the hippocampus and cortex tissues were determined by Western blot. The quantification of protein expression was determined by densitometry. Data are presented as mean ± SD (*n* ≥ 4). * *p* < 0.05 compared to sham group; # *p* < 0.05 compared to MCAO group. (**B**) A schematic summary of our main findings for the effects of both ICA and ICT on the acute cerebral ischemia–reperfusion injury is shown.

## Data Availability

The data presented in this study are available from the corresponding author upon reasonable request.

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
