# Peer review of "Bioactive Flavonoids Icaritin and Icariin Protect against Cerebral Ischemia–Reperfusion-Associated Apoptosis and Extracellular Matrix Accumulation in an Ischemic Stroke Mouse Model"

_biomedicines, 2021, doi:10.3390/biomedicines9111719_

Round 1

Reviewer 1 Report

The manuscript titled "Bioactive flavonoids icaritin and icariin protect against cerebral ischemia-reperfusion-associated apoptosis and extracellular matrix accumulation in an ischemic stroke mouse model" by Man-Chih and colleagues investigated the neuroprotective effects of Icaritin and icariin against ischemic stroke injury. The data are interesting and promising but I have the following concerns for this study. 1. The novelty is not high given that the neuroprotective effects of both Icaritin and icariin have been reported in stroke models, furthermore, the anti-inflammatory effects and anti-oxidative stress were also previously shown. 2. I'm concerned about their MCAO model; it seems to me that some brain tissues have no infarct in Figure 1c, which may result from an unsuccess of the MCAO stroke model. Furthermore, the authors did not report how to confirm the success of the stroke model. Therefore, I'm highly concerned about the rigor of the study. 3. Authors mentioned that the MCAO period was 60 min while they also mentioned it was 50min in the method part, which is confusing. 4. mNSS test was used for evaluating the neurological deficit. I'm concerned about this test as the cited reference used this test for rats rather than mice, and mNSS may not be a good test for the mice stroke model. The authors did not provide details about the test, for example, the diameters of the beam they used for the balance test. 5. The authors did not specify the route and time of drug treatments. 6. Authors should have citations about their dose selection in the method part.

Author Response

Reviewer 1

The manuscript titled "Bioactive flavonoids icaritin and icariin protect against cerebral ischemia-reperfusion-associated apoptosis and extracellular matrix accumulation in an ischemic stroke mouse model" by Man-Chih and colleagues investigated the neuroprotective effects of Icaritin and icariin against ischemic stroke injury. The data are interesting and promising but I have the following concerns for this study.

  1. The novelty is not high given that the neuroprotective effects of both Icaritin and icariin have been reported in stroke models, furthermore, the anti-inflammatory effects and anti-oxidative stress were also previously shown.

Our response:

We appreciate the reviewer's comment. We sincerely hope that the reviewer will understand that our research still has its novelty. Initially, we have carefully searched the information of both icariin (ICA) and icaritin (ICT) for brain ischemic stroke in PubMed. We didn’t find any information between ICT and ischemic brain injury, although there were several papers for effects of ICA on ischemic stroke in mice (Refs 23,24). Recently, a local literature, which was not listed in PubMed, has shown that ICT protects against ischemic brain injury in mice (Ref. 17). Although the anti-inflammatory and anti-oxidative effects of both ICA and ICT have been shown, the detailed effects and mechanisms of both ICT and ICA on acute cerebral ischemic stroke still remain to be clarified. In this study, we aimed to investigate and compare the effects and mechanisms of ICT and ICA on acute ischemic stroke. Our results demonstrated for the first time that both ICT and ICA pretreatment ameliorate the acute cerebral ischemia-reperfusion injury through the improvement in apoptotic neuronal cell death, ROS/RNS-induced lipid peroxidation, ECM accumulation, and EndMT-related fibrosis in the mouse brains (Figure 6B of this revised manuscript).

  1. I'm concerned about their MCAO model; it seems to me that some brain tissues have no infarct in Figure 1c, which may result from an unsuccess of the MCAO stroke model. Furthermore, the authors did not report how to confirm the success of the stroke model. Therefore, I'm highly concerned about the rigor of the study.

Our response:

We appreciate the reviewer's comment. A Laser Doppler (PeriFlux 4001, Perimed, Stockholm, Sweden) was used to monitor the cerebral blood flow for each mouse during MCAO as described in our previous studies (Refs. 28,29). The cerebral blood flow was reduced to less than 30% of the pre-ischemic value. We have added this information in the Methods of this revised manuscript. Moreover, we have revised the Figure 1C that the six photographs of the mouse cerebral infarct sections of each group (n=6) were shown.

  1. Authors mentioned that the MCAO period was 60 min while they also mentioned it was 50min in the method part, which is confusing.

Our response:

We appreciate the reviewer's comment. We have revised these descriptions in the Methods of this revised manuscript according to the suggestion of reviewer.

Drugs edaravone, ICA, and ICT were given by intraperitoneal injection (i.p.) before the focal cerebral ischemia for 1 hour, and then the procedure of MCAO surgery was carried out (for 50 min occlusion of the middle cerebral artery).

  1. mNSS test was used for evaluating the neurological deficit. I'm concerned about this test as the cited reference used this test for rats rather than mice, and mNSS may not be a good test for the mice stroke model. The authors did not provide details about the test, for example, the diameters of the beam they used for the balance test.

Our response:

We appreciate the reviewer's comment. We are sorry for this inadequate citation. We have revised the description and the cited reference in the Methods of this revised manuscript according to the suggestion of reviewer.

In this study, we used the modified mouse mNSS scoring to assess neurological injury in stroke mice as previously described by Li et al. (Ref. 30). It was scored on a scale of 0 to 14 (normal score, 0; maximal deficit score, 14), which was detected for the behavior of motor, sensory, reflex, and balance.  

  1. The authors did not specify the route and time of drug treatments.

Our response:

We appreciate the reviewer's comment. We have revised this issue in the Methods section of this revised manuscript according to the suggestion of reviewer.

Drugs edaravone, ICA, and ICT were given by intraperitoneal injection (i.p.) before the focal cerebral ischemia for 1 hour, and then the procedure of MCAO surgery was carried out. After 24 h of reperfusion, all mice were euthanized and underwent necropsy to observe the ischemic stroke signal networks.

  1. Authors should have citations about their dose selection in the method part.

Our response:

We appreciate the reviewer's comment. We have revised this issue in the Methods section of this revised manuscript according to the suggestion of reviewer.

Previously, a dosage of 60 mg/kg for ICA treatment has been found to significantly improve the ischemic stroke in an animal model [Ref 25]. Treatment with 60 mg/kg of ICT has also been shown to significantly ameliorate the high-fat diet-induced hepatic steatosis in an animal model [Ref 26].

In this study, the dosage of 60 mg/kg for both ICA and ICT was selected according to the previous studies (Refs 25,26) and our preliminary test.

Reviewer 2 Report

The manuscript from Chen et al. describes the neuroprotectant activities of icaritin (ICT) and icartin (ICA) against a middle cerebral artery occlusion mouse model. The results obtained by the authors demonstrate that those compounds reduced the infarct volume and neurological injury, as well as the oxidative and nitrosative stress impact, in parallel with a reduction in the lipid peroxidation levels and neural cell apoptosis.
The methods are well described, and the results are in general, well presented. However, an improvement of images from some figures is needed:
In Figure 1, image C needs to be represented in a bigger size. On the other hand, images f-j from cortex slices with DAPI staining require higher contrast to achieve better quality. Also, western blot images from figure 5 related to calveolin-1, e-NOS, and i-NOS expression levels in the hippocampus, and those western blot images of fibronectin and vimentin expression in both cortex and hippocampus require an improvement.

Finally, spelling mistakes need to be corrected, for example, those in lines 116 and 167: infarct instead of infract.

Author Response

Reviewer 2

The manuscript from Chen et al. describes the neuroprotectant activities of icaritin (ICT) and icartin (ICA) against a middle cerebral artery occlusion mouse model. The results obtained by the authors demonstrate that those compounds reduced the infarct volume and neurological injury, as well as the oxidative and nitrosative stress impact, in parallel with a reduction in the lipid peroxidation levels and neural cell apoptosis. The methods are well described, and the results are in general, well presented. However, an improvement of images from some figures is needed:

  1. In Figure 1, image C needs to be represented in a bigger size. On the other hand, images f-j from cortex slices with DAPI staining require higher contrast to achieve better quality.

Our response:

We appreciate the reviewer's comment. According to the suggestion of reviewer, we have revised these images with a bigger size and higher contrast in this revised manuscript.

  1. Also, western blot images from figure 5 related to calveolin-1, e-NOS, and i-NOS expression levels in the hippocampus, and those western blot images of fibronectin and vimentin expression in both cortex and hippocampus require an improvement.

Our response:

We appreciate the reviewer's comment. According to the suggestion of reviewer, we have revised these western blot images in this revised manuscript.

  1. Finally, spelling mistakes need to be corrected, for example, those in lines 116 and 167: infarct instead of infra

Our response:

We appreciate the reviewer's comment. We have carefully checked and revised the typos and grammatical errors of this manuscript according to the suggestion of reviewer.